# An evaluation of Mp1p antigen screening for talaromycosis in HIV-infected antiretroviral therapy-naïve population in Guangdong, China

**Dandan Gong**[1◉]**, Weiyin Lin**[1◉]**, Huihua Zhang**[1◉]**, Xu Ou**[1]**, Liya Li**[2]**, Pengle Guo**[1]**, Yaozu He**[1]**, Cong Liu**[1]**, Weiping Cai**[1]**, Xiaoping Tang**[2]**, Linghua Li**[1] *

**1** Infectious Disease Center, Guangzhou Eighth People's Hospital, Guangzhou Medical University, Guangzhou, Guangdong, China, **2** Research Institute, Guangzhou Eighth People's Hospital, Guangzhou Medical University, Guangzhou, Guangdong, China

◉ These authors contributed equally to this work.
* llheliza@126.com

**Data Availability Statement:** The authors confirm that all data underlying the findings are fully available without restriction. All relevant data are

## Abstract

### Background

Talaromycosis is one of the most common opportunistic infections in human immunodeficiency virus (HIV) infected patients. However, few researches have explored the prevalence in Southern China and fully assessed the value of the Mp1p antigen screening for the diagnosis of talaromycosis.

### Methodology/Principal findings

We performed a cross-sectional study of HIV-infected antiretroviral therapy (ART)-naïve adult patients who were seen in 2018 at Guangzhou Eighth People's Hospital, Guangzhou Medical University. Serum samples collected from all the 784 enrolled patients were tested for Mp1p antigen using double-antibody sandwich enzyme-linked immunosorbent assay. A culture of pathogen was conducted in 350 clinically suspected patients to confirm talaromycosis. The overall prevalence of talaromycosis based on the Mp1p antigen detection was 11.4% (89/784) and peaked at 32.2% (75/233) in patients with CD4$^+$ ≤50 Nr/μl. Logistic regression analysis found Mp1p antigen positive rate decreased with the increase in CD4$^+$ counts (OR 0.982, 95% CI 0.977–0.987, *P*<0.01). The optimal cut-off point of the CD4$^+$ count was 50 Nr/μl or less. Among the 350 patients received both fungal culture and Mp1p antigen detection, 95/350 (27.1%) patients were culture-positive for a *Talaromyces marneffei*, 75/350 (21.4%) patients were Mp1p antigen positive. The Mp1p antigen assay showed a good agreement to the culture of pathogen, and the sensitivity, specificity, positive predictive value, negative predictive value and kappa value was 71.6% (68/95), 97.3% (248/255), 90.7% (68/75), 90.2% (248/275), and 0.737, respectively. The screening accuracy of the Mp1p antigen assay in patients with CD4$^+$ counts of ≤50 Nr/μl was superior to that in those with higher CD4$^+$ counts.

within the paper and its Supporting Information files.

**Funding:** This work was supported by The National Key Research and Development Program of China (No.2022YFC2304800 to XPT) China National Health Development Research Center, http://www.nhei.cn/nhei/index.shtml; the National Natural Science Foundation of China (No.82072265 to LHL) National Natural Science Foundation of China, https://www.nsfc.gov.cn/; the Guangzhou Municipal Science and Technology Project (No. 20220020285 to LHL) Guangzhou Municipal Science and Technology Bureau, http://kjj.gz.gov.cn/; the Basic and Applied Basic Research Foundation of Guangdong Province (No. 2022A1515110546 to LHL) Department of Science and Technology of Guangdong Province, http://gdstc.gd.gov.cn/; the Medical Key Discipline Program of Guangzhou-Viral Infectious Diseases (2021–2023 to LHL, (no grant-number) Guangzhou Municipal Health Commission, http://wjw.gz.gov.cn/; the Guangzhou Municipal Science and Technology Project (No. 20220020276 to LHL) Guangzhou Municipal Science and Technology Bureau, http://kjj.gz.gov.cn/; the Guangzhou Municipal Science and Technology Project (No. 202201010874 to LHL) Guangzhou Municipal Science and Technology Bureau, http://kjj.gz.gov.cn/. The funders had no role in study design, data collection and analysis, decision to publish, or preparation of the manuscript.

**Competing interests:** The authors have declared that no competing interests exist.

## Conclusions/Significance

Mp1p antigen screening can be an effective tool for more efficient diagnosis of Talaromycosis, especially in HIV/AIDS patients with low $CD4^+$ counts. Future validation studies are needed.

## Author summary

*Talaromyces marneffei* (*T. marneffei*) is an important human immunodeficiency virus (HIV)-associated opportunistic pathogen in Southern China, and its infection is a leading cause of HIV-associated death. Properly and early antifungal treatment and timely antiretroviral therapy (ART) would reduce the mortality. Currently, the gold standard of diagnosis for talaromycosis is culture. The Mp1p antigen assay, the detection for a *T. marneffei*-specific protein, would be a reliable biomarker of *T. marneffei* infection in clinical practices. The $CD4^+$ counts are an indicator of human immune system. The less $CD4^+$ counts the weaker immune function and much possible the infection of opportunistic pathogen. In this study, we found that HIV patients with talaromycosis had a lower $CD4^+$ count and more advanced disease compared to those without. The prevalence of talaromycosis based on the Mp1p antigen detection was significantly higher in patients with low $CD4^+$ count than those with without. What's more, the Mp1p antigen screening was an effective tool to motivate the diagnosis for talaromycosis in patients with $CD4^+ \leq 50$ Nr/μl, and in turn to drive the initiation of antifungal treatment and ART, which would help the reduction of mortality and the improvement of treatment outcome.

## Introduction

Talaromycosis (formerly penicilliosis), caused by *Talaromyces marneffei* (*T. marneffei*), is a disseminated fungal infection that involves multiple organ systems and causes significant morbidity and mortality. Talaromycosis is one of the most common opportunistic infections (OIs) in human immunodeficiency virus (HIV) infected patients in endemic areas of Southeast Asia [1–3]. Even more alarming is the increasing number of talaromycosis cases reported in non-endemic areas, especially in HIV-infected patients with a low $CD4^+$ counts [4–6]. Recent study found talaromycosis has become the leading cause of HIV-associated death, with a high in-hospital mortality rate of up to 25 per 100 person-months [3]. Therefore, exploring the prevalence of talaromycosis is essential to improve the treatment strategy and reduce the mortality of people living with HIV [7,8].

Screening is recommended by World Health Organization (WHO) as the important component for guiding resources in a public health approach for identifying major OIs [9]. It has been reported that screening for cryptococcal antigen (CrAg) in the blood of patients with HIV, prior to the initiation of antiretroviral therapy (ART), can identify those at risk of developing cryptococcal meningitis (CM) [10–12]. However, screening for talaromycosis has not been implemented optimally over the years. A number of diagnostic methods for *T. marneffei* infection have been developed [13,14], but researches on assessing their diagnostic utility in screening for talaromycosis are insufficient. The gold standard for the diagnosis of *T. marneffei* infection is microbiological culture. But this is time-consuming, taking approximately 1 to 2 weeks to yield a positive result, making early clinical diagnosis and effective treatment difficult. The Mp1p antigen is a *T. marneffei*-specific protein which located throughout the cell wall of

*T. marneffei* yeast and is abundantly secreted, with its detection has the advantage of time saving, making it an ideal and specific target for the rapid and early diagnosis [14,15]. Mp1p antigen detection had been recommended in a global guideline on endemic mycoses which published in 2022 [16]. However, well-designed studies to fully assess its utility in screening for talaromycosis in real-world clinical practices among HIV-infected population are still lacking.

Southern China is one of the major endemic regions of *T. marneffei*, thus the prevention and control of talaromycosis are urgently needed. Unfortunately, the lack of screening remains a major obstacle to improve the treatment of *T. marneffei*-infected patients. To address the knowledge gaps, we performed this study to evaluate the prevalence of talaromycosis based on the Mp1p antigen detection and to assess the utility of Mp1p antigen screening among HIV-infected ART-naïve population in Guangdong, China.

## Methods

### Ethics statement

This study was approved by the Ethical Committee of the Guangzhou Eighth People's Hospital (GZ8H), Guangzhou Medical University (Approval No.202034167). All patients signed the informed consent.

### Study setting and population

The present study enrolled patients at Guangzhou Eighth People's Hospital in 2018 with the following design. Inclusion criteria were 1) confirmed HIV infection by HIV antibody or HIV nucleic acid detection, 2) age ≥18 years, 3) antiretroviral therapy-naïve, and 4) having complete data. Exclusion criteria were 1) pregnant or lactating women, and 2) patients who had received antifungal therapy before starting ART.

According to the results of Mp1p antigen detection, all subjects were split into two groups: the Mp1p positive group, including patients with the detectable Mp1p antigen in their serum sample, and the Mp1p negative group, including those with undetectable Mp1p antigen (Fig 1).

### Demographic data and clinical information collection

Data were collected from electronic medical records of enrollment, including demographic and clinical data. In addition, data from the case record form were independently double-entered to ensure data integrity.

An AIDS physician evaluated the stage of HIV/AIDS according to the WHO guidelines [17]. OIs were defined as those infections caused by conditional pathogens not harmful to healthy populations but to immunodeficient populations, including tuberculosis (TB), pneumocystis pneumonia (PCP), cytomegalovirus (CMV), candida albicans, etc.

### Mp1p antigen test in the serum samples

Serum samples were obtained within 3 days of enrollment. A double-antibody sandwich ELISA (DAS-ELISA) was used to detect the Mp1p antigen. Mp1p antigen testing kits were purchased from Wantai (Wantai Mp1p antigen ELISA; Wantai Biological Pharmacy Enterprise Co. Ltd, Beijing, China). As previously described, the Mp1p protein was produced using the *Pichia pastoris* expression system [15,18]. The assay was performed as follows. First, the serum samples were thawed at a room temperature of 25°C. Then, 20 μL of biotin reagent and 50 μL of serum specimen were added to the microplate, which was pre-coated with the rabbit

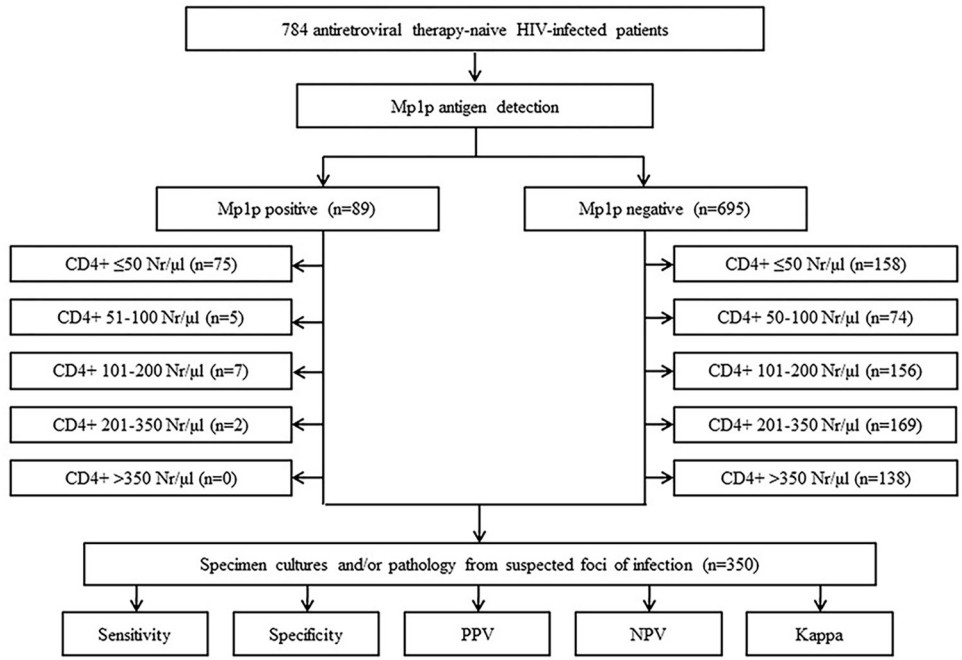

**Fig 1. Flow chart of the study.** PPV: positive predictive value, NPV: negative predictive value, Kappa: Cohen's kappa coefficient.

anti-Mp1p polyclonal antibody and incubated at 37˚C for 30 min. The plates were washed five times and dried, then 100 μL of biotinylated mouse anti-Mp1p monoclonal antibody was added, and the plates were incubated at 37˚C for 30 min. The plates were then washed five times, dried again, and incubated at 37˚C for 15 min, avoiding light, after adding chromogenic agent A (50μL) and chromogenic agent B (50μL) to each well. Finally, 50 μL of stopping solution was added to each well. The plates were examined in a microplate reader at 450 nm.

The cut-off value was 0.1 plus the average of negative control wells. Samples with a value greater than or equal to the cut-off were considered positive, and those below the cut-off value were negative.

## Specimen culture for talaromycosis

The gold standard for diagnosis of talaromycosis is specimen culture, including blood, bone marrow, bronchoalveolar lavage fluid (BALF) or other body fluid samples, and/or histopathological examination of clinically suspected foci of *T. marneffei* infection, according to the guidelines for the prevention and treatment of opportunistic infections [19]. The patients' blood was cultured following the study design. Invasive tests were performed only when the patient had a clinically suspected specific focus of infection and provided informed consent. The isolation of T. marneffei was conducted with the standard culture techniques, and the identification was based on the morphological and microscopic observation of the colonies, as previously reported [19–21].

## Statistical analysis

Continuous variables were expressed as the median and interquartile range (IQR), and categorical variables were expressed as numbers and percentages. Wilcoxon rank-sum and chi-squared tests were used for continuous and categorical variables, respectively. The utility of

baseline CD4$^+$ counts for classifying patients according to results of Mp1p antigen detection was examined using receiver operating characteristic (ROC) curve analysis and Youden's index (J) to determine the optimal cut-off point. Logistic regression analysis was used to identify the association between CD4$^+$ counts and Mp1p antigen status. The sensitivity, specificity, positive predictive value (PPV), negative predictive value (NPV), and kappa value were calculated to evaluate the coincidence between Mp1p antigen detection and microbiological culture. All analyses were performed with the statistical software SPSS 25. A *P*-value <0.05 was considered statistically significant.

## Results

### Demographic characteristics of patients

Table 1 shows the baseline characteristics in this study. Among the 784 enrolled patients, the median age was 37.0 years, 87.8% (688/784) were male, and 78.6% (616/784) were infected with HIV by sexual intercourse. The median CD4$^+$ count was 151 Nr/μl (IQR, 35–289 Nr/μl), and the median CD4$^+$/CD8$^+$ ratio was 0.20 (IQR, 0.09–0.36). Totally 89 of the 784 (11.4%) patients had detectable Mp1p antigen in serum samples. Compared to the Mp1p negative group, the Mp1p positive group present with more advanced HIV disease, characterized by

**Table 1. Baseline Characteristics of 784 Participants Undergoing Mp1p Antigen Screening Stratified by Mp1p Antigen Status.**

| Characteristics | Total (n = 784) | Mp1p positive (n = 89) | Mp1p negative (n = 695) | *P value* |
|---|---|---|---|---|
| **Age** | 37(27,49) | 37(29,46) | 37(27,49) | 0.591[a] |
| **Male** | 688(87.8%) | 80(89. 9%) | 608(87.5%) | 0.515[b] |
| **Source of patients** | | | | *P<0.001* [b] |
| Inpatient | 354(45.2%) | 74(83.2%) | 280(40.3%) | |
| Outpatient | 430(54.8%) | 15(16.8%) | 415(59.7%) | |
| **Route of HIV infection** | | | | 0.800 [b] |
| Sex | 616(78.6%) | 68(76.4%) | 548(78.9%) | |
| Blood | 25(3.2%) | 2(2.3%) | 23(3.3%) | |
| Others | 70(8.9%) | 10(11.2%) | 60(8.6%) | |
| Unknown | 73(9.3%) | 9(10.1%) | 64(9.2%) | |
| **CD4$^+$ Count (Nr/μl)** | 151(35,289) | 12(6,38) | 179(61,315) | *P<0.001*[a] |
| ≤50 | 233(29.7%) | 75(84.3%) | 158(22.7%) | *P<0.001*[b] |
| 51–100 | 79(10.1%) | 5(5.6%) | 74(10.7%) | |
| 101–200 | 163(20.8%) | 7(7.9%) | 156(22.4%) | |
| 201–350 | 171(21.8%) | 2(2.2%) | 169(24.3%) | |
| >350 | 138(17.6%) | 0(0.0%) | 138(19.9%) | |
| **CD4$^+$/CD8$^+$ ratio** | 0.20(0.09,0.36) | 0.05(0.03,0.10) | 0.22(0.10,0.38) | *P<0.01*[a] |
| **Stage of HIV/AIDS** | | | | *P<0.001*[b] |
| I/II | 476(60.7%) | 13(14.6%) | 463(66.6%) | |
| III | 160(20.4%) | 6(6.7%) | 154(22.2%) | |
| IV | 148(18.9%) | 70(78.7%) | 78(11.2%) | |
| **OIs infection** | 315(40.2%) | 77(86.5%) | 238(34.2%) | *P<0.001*[b] |
| **HBV infection** | 88(11.2%) | 15(16.9%) | 73(10.5%) | 0.074 |
| **HCV infection** | 29(3.7%) | 1(1.1%) | 28(4.0%) | 0.285[c] |

a Mann-Whitney *U* test

b Pearson test, and

c Fisher's exact test.

significantly lower CD4$^+$ counts (12 Nr/µl [IQR, 6–38 Nr/µl] vs. 179 Nr/µl [IQR, 61–315 Nr/µl], $P<0.001$), lower CD4$^+$/CD8$^+$ ratio (0.05 [IQR, 0.03–0.10] vs. 0.22 [IQR, 0.10–0.38], $P<0.001$), higher proportion of HIV/AIDS stage IV (78.7% vs. 11. 2%, $P<0.001$), higher proportion of other known OIs (86.5% vs. 34.2%, $P<0.001$) and higher risk of hospitalization (83.2% vs. 40.3%, $P<0.001$).

## Prevalence of talaromycosis based on the Mp1p antigen detection in patients with different CD4$^+$ count levels

We stratified the CD4$^+$ count according to the cut-off values commonly used in clinical practice by ≤50 Nr/µl, 51–100 Nr/µl, 101–200 Nr/µl, 201–350 Nr/µl, and >350 Nr/µl. The positive rate of Mp1p antigen detection peaked at 32.2% (75/233) in patients with CD4$^+$ ≤50 Nr/µl and gradually decreased in the following four groups with higher CD4$^+$ count, which were 6.3% (5/79), 4.3% (7/163), 1.2% (2/171), and 0.0%, respectively (Fig 2, $P$ value for trend <0.001). To identify the association between CD4$^+$ counts and Mp1p antigen status, we perform a logistic regression analysis and found Mp1p antigen positive rate decreased with the increase in CD4$^+$ counts (OR 0.982, 95% CI 0.977–0.987, $P<0.01$). ROC curve analysis revealed the ability of CD4+ count to discriminate between patients who were Mp1p antigen positive or negative with the area under ROC curve [AUC] = 0.870 (95% CI, 0.835–0.906, $P<0.001$) (Fig 3). Analyses to determine an optimal CD4$^+$ count cutoff to classify patients with or without Mp1p antigen using the Youden index ($J$) indicated the optimal cutoff is 50 Nr/µl or less, yielding a sensitivity of 84.3% and specificity of 77.7% (Table 2). Therefore, the prevalence of talaromycosis based on the Mp1p antigen detection in serum samples among the overall patients, those with CD4$^+$ ≤50 Nr/µl and CD4$^+$ >50 Nr/µl was 11.4% (89/784), 32.2% (75/233) and 2.5% (14/551), respectively. Patients with CD4$^+$ ≤50 Nr/µl had 18 times higher chance of Mp1p antigen positive than those with CD4$^+$ >50 Nr/µl. Similar results were found in subgroup analyses according to source of patients (inpatient department or outpatient department).

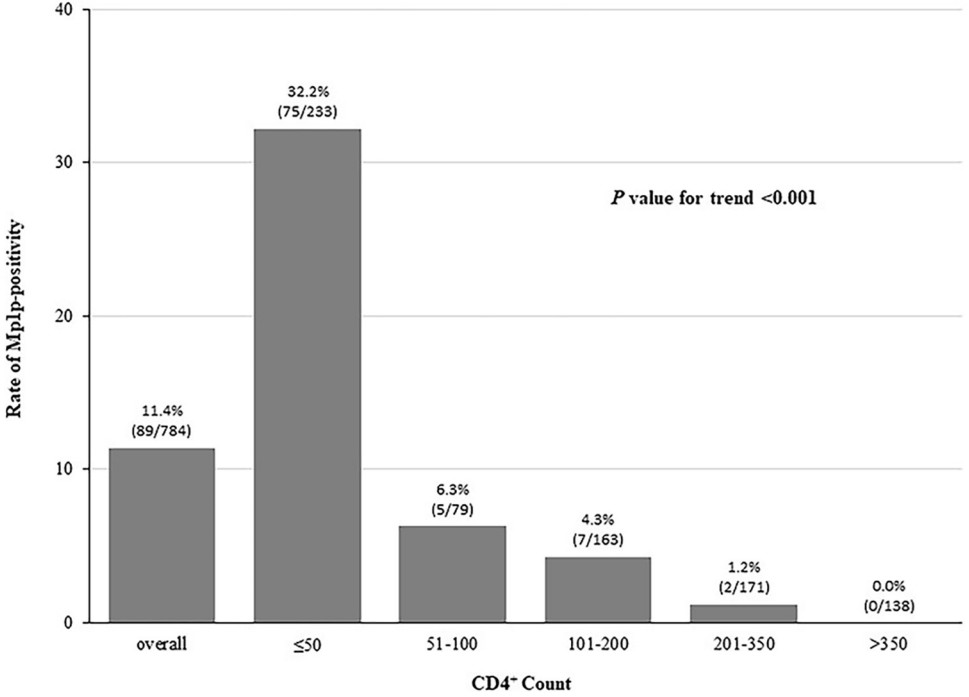

**Fig 2. Rates of the Mp1p-positive case in different CD4$^+$ count level.**

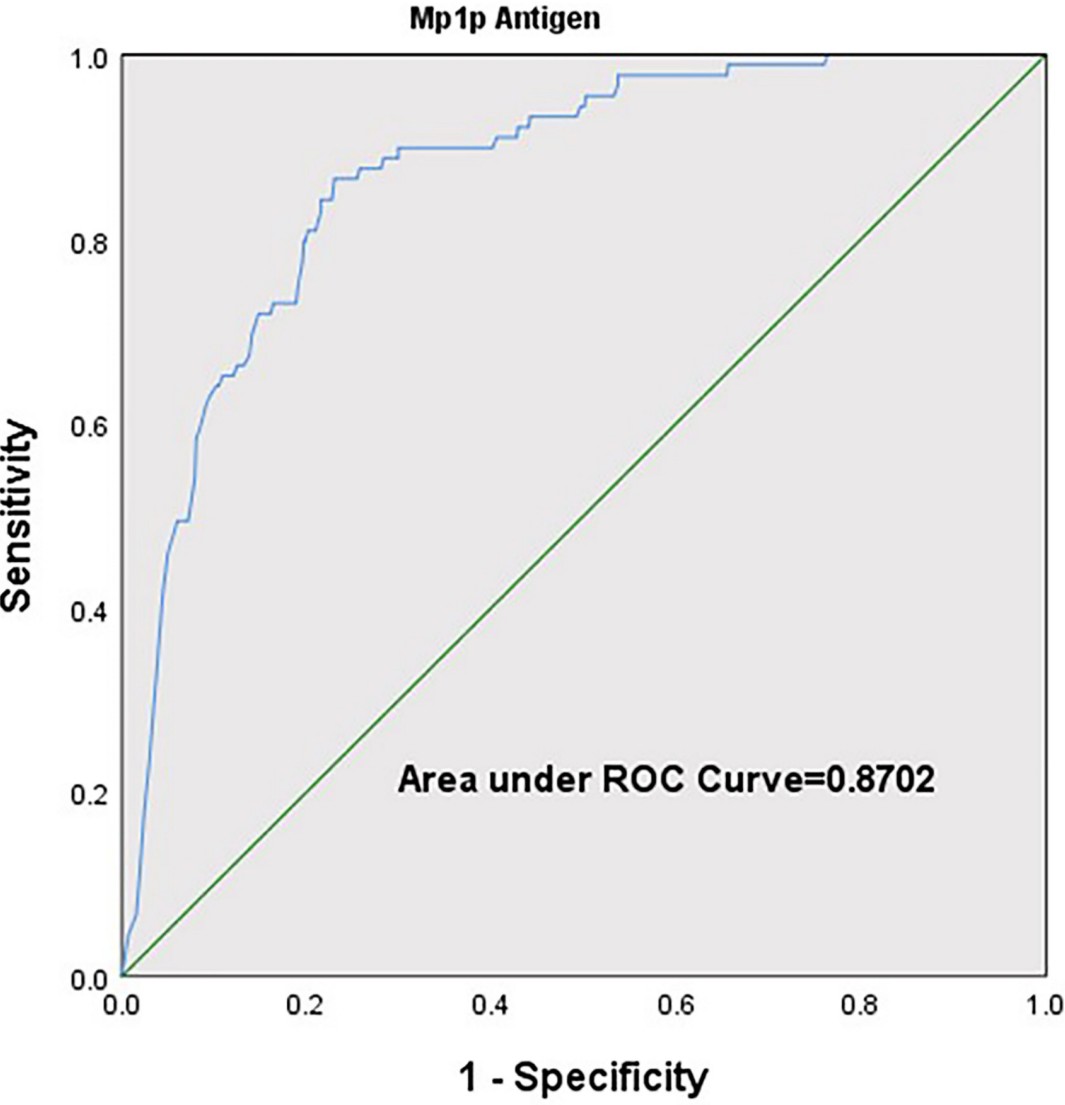

**Fig 3. Receiver operating characteristic (ROC) curves.**

### Fungal culture and Mp1p antigen detection

A culture of pathogen was conducted in 350 clinically suspected patients, over 99% of whom came from the inpatient department, to confirm talaromycosis. The CD4$^+$ count and the prevalence of Mp1p antigen among these patients were shown in Fig 4. The CD4$^+$ counts of these 350 patients showed the similar trend with those of the total of 784 participants.

**Table 2. Cut-point analysis using Youden's J Index.**

| CD4$^+$ Count (Nr/µl) | Sensitivity | Specificity | Youden's $J$ |
|---|---|---|---|
| ≤50 | 0.842697 | 0.776978 | 0.619675 |
| ≤100 | 0.898876 | 0.666187 | 0.565063 |
| ≤200 | 0.977528 | 0.441727 | 0.419255 |
| ≤350 | 1.000000 | 0.198561 | 0.198561 |

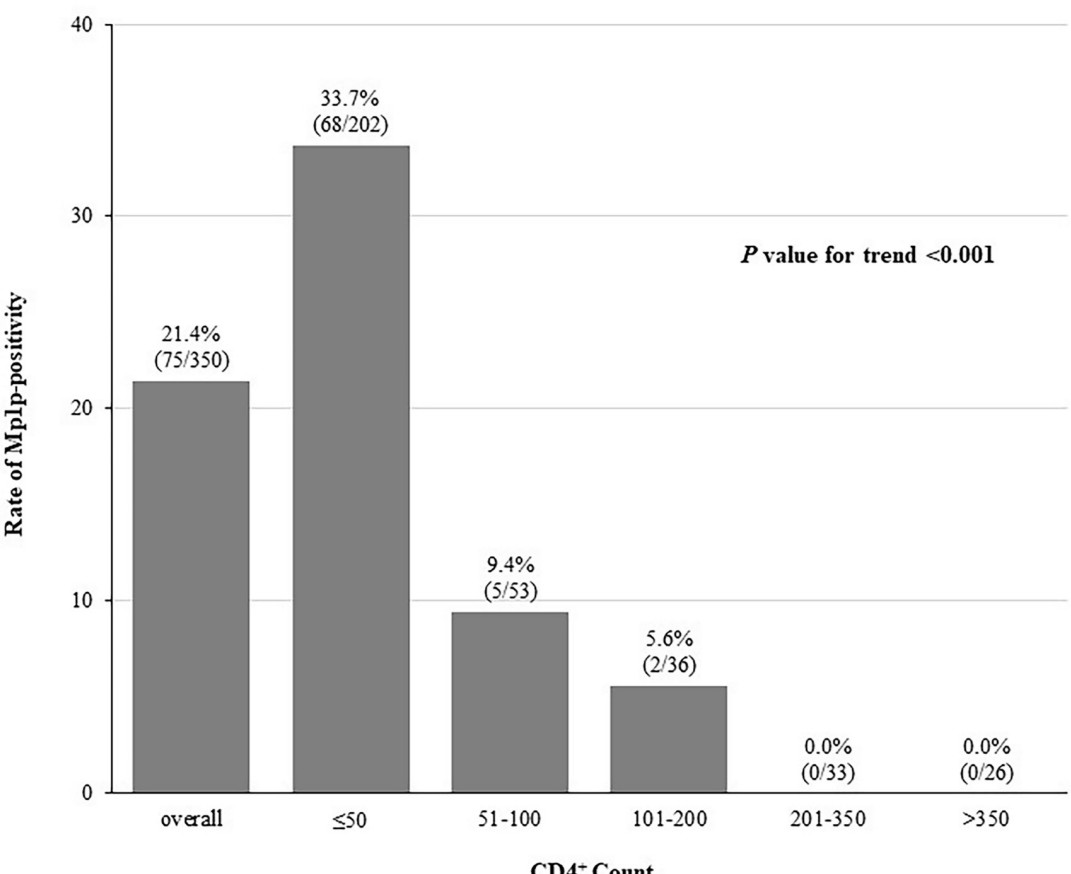

**Fig 4. Rates of the Mp1p-positive case in different CD4$^+$ count level among patients received specimen culture and/or histopathological examination.**

*T. marneffei* was isolated from the clinical specimens of blood, bone narrow, bronchial alveolar lavage and others. Among the 350 patients receiving both fungal culture and Mp1p antigen detection, 95/350 (27.1%) patients were culture-positive for a *T. marneffei*, 75/350 (21.4%) patients were Mp1p antigen positive. The Mp1p antigen assay showed a good agreement to the culture of pathogen. The sensitivity, specificity, PPV, NPV, and kappa value were 71.6% (68/95), 97.3% (248/255), 90.7% (68/75), 90.2% (248/275), and 0.737, respectively. The screening accuracy of the Mp1p antigen assay in patients with CD4$^+$ counts of ≤50 Nr/μl was superior to that in those with higher CD4$^+$ counts, shown in Table 3 with a sensitivity of 73.6%, a specificity of 96.5%, a PPV of 94.1% and a NPV of 82.8% and a higher Kappa value (0.720 vs. 0.509).

**Table 3. Screening efficacy of Mp1p antigen in the various CD4$^+$ count levels.**

| | | Culture (-) | Culture (+) | Sensitivity | Specificity | PPV | NPV | Kappa |
|---|---|---|---|---|---|---|---|---|
| Total patients (n = 350) | Mp1p (-) | 248 | 27 | 71.6% | 97.3% | 90.7% | 90.2% | 0.737 |
| | Mp1p (+) | 7 | 68 | | | | | |
| CD4$^+$ ≤50 Nr/μl (n = 202) | Mp1p (-) | 111 | 23 | 73.6% | 96.5% | 94.1% | 82.8% | 0.720 |
| | Mp1p (+) | 4 | 64 | | | | | |
| CD4$^+$ >50 Nr/μl (n = 148) | Mp1p (-) | 137 | 4 | 50.0% | 97.9% | 57.1% | 97.2% | 0.509 |
| | Mp1p (+) | 3 | 4 | | | | | |

## Discussion

This study evaluated the prevalence of talaromycosis and assessed the value of Mp1p antigen screening in HIV infected ART-naïve population. We found that performing the Mp1p antigen screening, especially in high-risk population with $CD4^+$ ≤50 Nr/μl, might contribute to the improvement of early diagnosis and timely treatment of talaromycosis.

The $CD4^+$ cell count is the best indicator of disease stage that was associated with the risk of OIs [9]. Talaromycosis, as reported previously, is a common HIV-associated OI that is endemic in Southeast Asia and Southern China [3, 22, 23]. We observed a significant difference in the $CD4^+$ count at baseline between the results of Mp1p antigen detection, with the consistent observations of high ratios of the OIs and advanced HIV disease in the Mp1p antigen positive group. More than 80% of the Mp1p antigen positive patients had a $CD4^+$ count of less than 50 Nr/μl, and the Mp1p-positive rate peaked at this subgroup. Logistic regression analysis in present study further showed that decreasing $CD4^+$ count was positively associated with the presence of Mp1p antigen. The above data supported a high prevalence of talaromycosis in HIV infected patients, especially in those with low $CD4^+$ count. Our study was consistent with previous researches. Qin et al. evaluated the prevalence of talaromycosis and found that patients with a low $CD4^+$ count of less than 200 Nr/μl had a higher risk of *T. marneffei* infection [24]. Wang et al. reports that patients with a $CD4^+$ count of less than 50 Nr/μl were prone to various OIs, including talaromycosis [6]. A previous research conducted in Guangdong during 2011–2017 found that 91.9% of HIV infected patients with talaromycosis had a $CD4^+$ count less than 50 Nr/μl [25]. These findings above are not surprising since a lower $CD4^+$ count means weaker immunity, and people presenting with advanced HIV disease are more vulnerable to *T. marneffei* infection. What's more, in our study, ROC curve analysis proved the $CD4^+$ count of 50 Nr/μl or less was the optimal cut-off point to discriminate the presence or absence of the Mp1p antigen with an AUC of 0.870. Although it was not possible to completely identify the presence of Mp1p antigen using a single $CD4^+$ count cutoff. This screening tool would benefit for the outpatients without systematic and comprehensive physical and pathological examination in the regular follow-up to diagnose talaromycosis early and rapidly. In addition, performing the Mp1p screening among the patients without obvious symptoms might contribute to the rapid diagnosis at the early stage of talarymocosis.

Many previous researches had demonstrated that the Mp1p antigen could be a promising biomarker of talaromycosis. The Mp1p antigen detection has been shown an excellent specificity, between 90.0% and 99.6%, and superior sensitivity and time to diagnosis compared to standard blood culture [15]. In the present study, the Mp1p antigen was confirmed by specimen culture in 350 clinically suspected patients, observing a substantial agreement between Mp1p antigen detection and the gold standard. This result is consistent with the researches above. Compared to specimen culture, Mp1p antigen detection is time-saving, inexpensive and does not require sophisticated equipment. What's more, our data showed this agreement in patients with $CD4^+$ counts of ≤50 Nr/μl (Kappa = 0.720) was superior to that in those with higher $CD4^+$ counts (Kappa = 0.509). The better diagnostic accuracy of the Mp1p antigen was supported in the subgroup with low $CD4^+$ count, with a sensitivity of 73.6%, a specificity of 96.5%, a PPV of 94.1% and a NPV of 82.8%. Therefore, the Mp1p detection in our study was reliable in screening for talaromycosis in ART-naïve HIV infected patients. The high accuracy in individuals with $CD4^+$ ≤50 Nr/μl indicated that it is worth conducting the Mp1p antigen screening in this population presenting with advanced HIV disease. This screening strategy would contribute to the early diagnosis of talaromycosis, driving the initiation of timely and proper antifungal treatment, and to reduce the mortality.

Our research should be interpreted in light of some limitations. First, this is a single-center study, the prevalence of talaromycosis is representative only of the Southern China. Second, the CD4$^+$ count was unevenly distribution among enrolled patients, especially the number of patients with CD4+ count of 51–100 Nr/μl was relatively insufficient, which might underestimate the prevalence of talaromycosis in this subgroup. Third, as a cross-sectional study, clinical details such as some comprehensive physical and pathological examination in outpatients might be incomplete. Finally, assessing the value of Mp1p antigen screening by specimen culture was conducted in 350/784 (44.6%) clinically suspected patients, over 99% came from the inpatient department, only one of them came from outpatient department. However, it is not realistic for all HIV infected patients to receive specimen culture, especially given that it takes time, is expensive, and requires invasive methods to obtain samples such as bone marrow or bronchoalveolar lavage fluid. Therefore, multicenter, large sample size and longitudinal study with long-term follow up are needed to explore the prevalence of talaromycosis based on the Mp1p antigen detection in different endemic regions and to fully assess value of Mp1p antigen screening on cost-effective and clinical outcome.

In conclusion, Mp1p antigen screening can be an effective tool for more efficient diagnosis of talaromycosis, especially in HIV/AIDS patients with low CD4$^+$ counts. Future validation studies are needed.

## Supporting information

**S1 Table. Data of the research.**
(PDF)

## Acknowledgments

We thank all the patients who took part in this study and provided samples to support the scientific research.

## Author Contributions

**Formal analysis:** Dandan Gong.

**Funding acquisition:** Xiaoping Tang, Linghua Li.

**Project administration:** Linghua Li.

**Resources:** Xu Ou, Liya Li, Pengle Guo, Yaozu He, Cong Liu.

**Supervision:** Weiping Cai, Xiaoping Tang, Linghua Li.

**Writing – original draft:** Dandan Gong.

**Writing – review & editing:** Weiyin Lin, Huihua Zhang, Linghua Li.

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
