## [Decision Letter · Decision Letter 0]

28 Aug 2023

Dear Dr. Li,

Thank you very much for submitting your manuscript "An evaluation of Mp1p antigen screening for talaromycosis in HIV-infected antiretroviral therapy-naïve population in Guangdong, China" for consideration at PLOS Neglected Tropical Diseases. As with all papers reviewed by the journal, your manuscript was reviewed by members of the editorial board and by several independent reviewers. In light of the reviews (below this email), we would like to invite the resubmission of a significantly-revised version that takes into account the reviewers' comments. Here goes a comment placed by the handling editor:

Dear authors,

The manuscript titled "An evaluation of Mp1p antigen screening for talaromycosis in HIV-infected antiretroviral therapy-naïve population in Guangdong, China" was reviewed by two expert peers. Please review the comments of each of the reviewers and modify the manuscript taking these comments into account.

We cannot make any decision about publication until we have seen the revised manuscript and your response to the reviewers' comments. Your revised manuscript is also likely to be sent to reviewers for further evaluation.

Sincerely,

Angel Gonzalez, Ph.D.

Academic Editor

Marcio Rodrigues

Section Editor

Dear authors,

The manuscript titled "An evaluation of Mp1p antigen screening for talaromycosis in HIV-infected antiretroviral therapy-naïve population in Guangdong, China" was reviewed by two expert peers. Please review the comments of each of the reviewers and modify the manuscript taking these comments into account.

Reviewer's Responses to Questions

**Key Review Criteria Required for Acceptance?**

**Methods**

-Are the objectives of the study clearly articulated with a clear testable hypothesis stated?

-Is the study design appropriate to address the stated objectives?

-Is the population clearly described and appropriate for the hypothesis being tested?

-Is the sample size sufficient to ensure adequate power to address the hypothesis being tested?

-Were correct statistical analysis used to support conclusions?

-Are there concerns about ethical or regulatory requirements being met?

Reviewer #1: No, the objectives of the study weren't clearly articulated with a clear testable hypothesis stated.

Yes, the study design was appropriate to address the stated objectives.

Yes, the population was clearly described and appropriate for the hypothesis being tested.

Yes, the sample size was sufficient to ensure adequate power to address the hypothesis being tested.

Yes, statistical analysis was correct used to support conclusions.

No, there aren't concerns about ethical or regulatory requirements being met

Reviewer #2: The methods for screening patients with Mp1p antigen testing followed by culture testing are sufficient to investigate the efficacy of the antigen test and support the conclusions in the manuscript. Ultimately, these data will benefit from a larger sample size, but the sample size in the present study appears sufficient to support the claims.

**Results**

-Does the analysis presented match the analysis plan?

-Are the results clearly and completely presented?

-Are the figures (Tables, Images) of sufficient quality for clarity?

Reviewer #1: Yes, the analysis presented match the analysis plan.

Yes, the results were clearly and completely presented.

No, the figures weren't of sufficient quality for clarity.

Yes, the Tables were of sufficient quality for clarity. 

There is not any pictures.

Reviewer #2: The results of the study are clear, if not ground-breaking. The authors show that Mp1p antigen screening is effective for detection in Talaromycosis in HIV/AIDS patients in Southern China. In many cases, the text could be written in a way that would more clearly convey the findings, as it is often unclear which association the researchers are attempting to make. For example, the text in lines 233-241 could be interpreted as Talaromycosis causing decreased CD4 counts in patients, rather than low CD4 counts leaving patients susceptible to Talaromycosis. This should be clarified throughout the text.

**Conclusions**

-Are the conclusions supported by the data presented?

-Are the limitations of analysis clearly described?

-Do the authors discuss how these data can be helpful to advance our understanding of the topic under study?

-Is public health relevance addressed?

Reviewer #1: No, the conclusions weren't supported by the data presented.

No, the limitations of analysis weren't clearly described.

Yes, the authors discussed how these data can be helpful to advance our understanding of the topic under study.

Yes, public health was relevance addressed.

Reviewer #2: The basic conclusion of the paper is that Mp1p antigen screening can be an effective tool for more efficient diagnosis of Talaromycosis, especially in HIV/AIDS patients with low CD4 counts. The data suggest this is a sound conclusion. Further discussion of the benefits of antigen testing would be useful. For example, how cost effective is this antigen test? How much time does it save, practically, compared to culture testing? Some data to this effect would bolster the authors' claims that antigen testing would be an effective strategy to implement earlier antifungal treatment in patients with low CD4 counts.

**Editorial and Data Presentation Modifications?**

Reviewer #1: ● The aim was clear.

● Abstract (291 words) was clear what the study found and how they did it

● The title “An evaluation of Mp1p antigen screening for talaromycosis in HIV-infected antiretroviral therapy-naïve population in Guangdong, China” was informative and relevant.

● Author Summary was was clear and understandable, and the author summarized all the things that should be stated.

● The references were: 26 references (17 new and 9 old references).

● The references must come from current scientific journals (c. 80% published in the last 10 years)

● Referenced were correctly.

● Appropriate were key studies included.

Reviewer #2: Editing for grammar throughout the manuscript will be crucial. 

Terms like PPV, NPV, kappa, and OI should be defined where they are first used.

In the introduction, the authors state that the mortality rate of T. marneffei in HIV patients is ~33%, a figure which comes from a small study by Son et al in 2014, but it seems more recent estimates, as in Jiang 2019 (Clinical Microbiology and Infection) may be more accurate.

Line 164 “The cut-off value was 0.1” needs units.

**Summary and General Comments**

Reviewer #1: Introduction / background

● Introduction wasn’t clear what is known about this topic, please add one page more about the topic with new references.

● The research questions were clearly outlined.

● The research questions were justified given what is known about the topic.

Methods: Experimental:

● The process of subject was selection clear.

● The variables were defined and measured appropriately.

● The study methods were valid and reliable.

● The sample size was very suitable for such a study.

● Great job from the authors for having an Ethics statement.

● There were enough detail in order to replicate the study.

● There weren't enough details on the method of Specimen culture for talaromycosis.

Results:

● The data was presented in an appropriate way.

● Tables were relevant and clearly presented.

● Figures weren’t relevant and clearly presented.

● The units, rounding, and number of decimals were appropriate.

● Titles, columns, and rows were labelled correctly and clearly.

● Categories were grouped appropriately.

● The text in the results was added to the data.

● I’m clear about what is a statistically significant result.

● I’m clear about what is a practically meaningful result.

● I didn't see any pictures that were crisp and beautifully done.

Discussion and Conclusions: 

● The results were discussed from multiple angles and placed into context without being overinterpreted.

● The conclusions weren’t to answer the aims of the study.

● The conclusions weren't supported by references or results.

● The limitations of the study give opportunities to make future research.

Reviewer #2: This study makes a reasonable case that implementation of Mp1p antigen testing is a sensitive and reliable assay for Talaromycosis in HIV/AIDS patients and could be used for early screening in endemic areas to get patients antifungal treatment more efficiently. These data are potentially informative and useful for this specific niche. Ultimately, the conclusions would benefit from additional data or discussion of the practicality of implementing this screening test.

PLOS authors have the option to publish the peer review history of their article (what does this mean?). If published, this will include your full peer review and any attached files.

Reviewer #1: No

Reviewer #2: No
---

## [Editor Report · Decision Letter 1]

8 Nov 2023

Dear PhD. Li,

We are pleased to inform you that your manuscript 'An evaluation of Mp1p antigen screening for talaromycosis in HIV-infected antiretroviral therapy-naïve population in Guangdong, China' has been provisionally accepted for publication in PLOS Neglected Tropical Diseases.

Best regards,

Marcio L Rodrigues

Section Editor

Marcio Rodrigues

Section Editor

Line 176, in the introduction section, please change "motility" for "mortality".

---

## [Editor Report · Acceptance letter]

20 Nov 2023

Dear PhD. Li,

We are delighted to inform you that your manuscript, "An evaluation of Mp1p antigen screening for talaromycosis in HIV-infected antiretroviral therapy-naïve population in Guangdong, China," has been formally accepted for publication in PLOS Neglected Tropical Diseases.

Best regards,

Shaden Kamhawi

co-Editor-in-Chief

Paul Brindley

co-Editor-in-Chief
